# Long Period Grating Mach–Zehnder Interferometer Based Immunosensor with Temperature and Bulk Refractive Index Compensation

**DOI:** 10.3390/bios12121099

**Published:** 2022-11-30

**Authors:** Peizhou Wu, Liangliang Liu, Stephen P. Morgan, Ricardo Correia, Serhiy Korposh

**Affiliations:** Optics and Photonics Research Group, University of Nottingham, University Park, Nottingham NG7 2RD, UK

**Keywords:** long period grating, Mach–Zehnder interferometer, biosensor, IgM, bulk refractive index

## Abstract

A long period grating Mach–Zehnder interferometer (LPGMZI) that consists of two identical long period gratings (LPGs) in a single fibre was developed to measure immunoglobulin M (IgM). The measured spectrum has fringes due to the interference between the core mode and cladding mode. This immunosensor inherits the advantages of an LPG and has the potential to compensate for unwanted signal changes due to bulk refractive index (RI) and temperature fluctuations by analysing interference fringes and their envelope. The external RI was measured from 1.3384 to 1.3670 in two different cases: (i) only the connecting section between the two LPGs is immersed or (ii) the whole LPGMZI is immersed. The fringes shift with an external RI in both scenarios, whereas the envelope stays still in case (i) or shifts at the same rate as the fringes in case (ii). The LPGMZI was also characterised at different temperatures between 25 °C and 30 °C by placing the whole LPGMZI in a water bath. The fringes and envelope shift at the same rate with temperature. The LPGMZI platform was then used to create an IgM immunosensor. The connecting section between the two LPGs was functionalised with anti-IgM and immersed into solutions with IgM concentrations from 20 μg/mL to 320 μg/mL. The fringes shift with IgM concentration and the envelope remains static. The results from this work show that LPGMZI has the potential to compensate for the temperature and bulk RI fluctuations and perform as a portable biosensor platform.

## 1. Introduction

A long period grating (LPG) is currently a widely used platform for developing chemical sensors and biosensors [1,2]. The properties of having a light weight, small size, and immunity to electromagnetic interference make an LPG an ideal platform for designing point-of-care (POC) portable biosensors. These sensors provide more labour saving and a faster sensing of biomolecules than ELISA since they do not require a labelling step [3]. LPG-based biosensors detect the change in the refractive index (RI) on the surface upon molecular binding. The main sources of cross-sensitivities to this include temperature fluctuations and changes in an external medium’s RI (bulk RI) [4,5,6]. Many efforts have been made to overcome these problems. For example, a plasmonic sensor with gold film deposited on the surface of tilted fibre Bragg grating (FBG) can discriminate the surface RI and bulk RI [7]. The same function can be implemented using an LPG with a micro-cavity in the middle [8]. A core-offset Mach–Zehnder interferometer fabricated in series with an FBG can implement a simultaneous monitoring of temperature during RI measurement [9], as does an LPG fabricated in series with an FBG [10].

The long period grating Mach–Zehnder interferometer (LPGMZI) described in this work consists of two identical LPGs in a single fibre. As shown in Figure 1, two identical LPGs are fabricated in series and light propagates along the optical fibre from left to right. LPG1 couples light into a cladding mode; the cladding-mode light propagates along the connecting section and is coupled back into the core by LPG2, where it interferes with light that has remained within the core [11]. This interference pattern is analysed on a spectrometer.

The transmission spectrum has fringes due to the interference between the core mode and cladding modes [11], which has the potential to compensate for unwanted signal changes caused by bulk RI changes and temperature fluctuations.

The interference pattern in the transmission spectrum of an LPGMZI changes with temperature, and a uniform change over the LPGMZI results in the fringes and envelope shifting at the same rate [12]. The interference pattern is also sensitive to external RIs. The fringes and envelope shift when the whole LPGMZI is immersed into solutions with different RIs [13]. In comparison to other methods [7,8,9,10], the LPGMZI is a more elegant structure used to compensate for both bulk RI and temperature changes with a single calibration step, a simpler set-up, and a relatively simpler fabrication method.

Chemical sensors and biosensors are developed by depositing functional films on an LPGMZI [14,15,16]. For example, an ammonia (NH_3_) sensor is designed by depositing graphene film on the connecting section of LPGMZI [14], which adsorbs NH_3_ to change its RI. Similarly, a zeolitic imidazole framework (ZIF-8) film is deposited on an LPGMZI to detect volatile organic compounds (VOCs) such as methanol and toluene [15]. A Mach–Zehnder interferometer can also be made by cascading two identical chirped LPGs with a separation between each other [16]. This was used to create a biosensor to detect *Escherichia coli (E. coli)* bacteria by depositing bioreceptors on the connecting section [16]. In all of the previous literature about LPGMZI, the fringes were analyzed, but the fringes’ envelope was not investigated quantitatively.

In this work, an LPGMZI was fabricated and characterised. The fringes and envelope were measured and analysed to provide a more thorough investigation of LPGMZI and more robust sensing. In the temperature measurement, the temperature over the whole LPGMZI changes uniformly. The external RI measurement was carried out in two scenarios: (i) the whole LPGMZI or (ii) only the connecting section is immersed into solutions with different RIs. An LPGMZI immunosensor used to detect immunoglobulin M (IgM) was developed.

## 2. Methodology

### 2.1. Experimental Set-Ups and Signal Processing

LPGs were fabricated in boron–germanium co-doped photosensitive fibre (Fibrecore PS750, Southampton, UK) using a 266 nm laser and an amplitude mask with a period of 113.5 μm (Suzhou Sunshine Laser Technology, Suzhou, China). The two LPGs were 2 cm in length with 6 cm separation. The number of fringes in the transmission spectrum increases with the length of separation [11]. A 6 cm separation was chosen to ensure that the envelope and higher-frequency fringes are distinguishable from each other in the frequency domain.

Two experimental set-ups are shown in Figure 2. Figure 2a is the set-up for external RI measurements and IgM measurements. Solutions with different RIs were prepared by dissolving sodium chloride into deionised (DI) water. A higher concentration of sodium chloride leads to a higher RI. The RIs of solutions were determined by a commercial refractometer (Reichert 13940000, Depew, NY, USA). The ends of the optical fibre were connected to a halogen tungsten light source (Ocean Optics, HL-2000, Orlando, FL, USA) and spectrometer (Ocean Optics, HR4000, Orlando, FL, USA), respectively. The LPGMZI was placed in a custom-made 3-section PLA-plastic fibre bath (middle section 4.5 cm (L) × 0.5 cm (W) × 0.3 cm (H), LPG sections 2.75 cm (L) × 0.5 cm (W) × 0.3 cm (H)). The connecting section was placed in the middle section of the bath. The two LPGs were located either side and were connected through small crevices of 0.3 mm, which ensures that the solution does not leak between sections due to the hydrophobicity of the bath material. Figure 2b shows the experimental set-up of temperature measurement. The whole LPGMZI was placed in a water heating bath (VDB05EU, Grant Instruments, Shepreth, UK, 30 cm (L) × 15 cm (W) × 15 cm (H)) that allows the temperature to be changed homogeneously. The total cost of the set-up was approximately GBP 5000, although this would be lower if manufactured in scale.

In the external RI measurements, the responses of envelope and fringes were investigated in two cases: (i) only the connecting section was immersed into different RI solutions and (ii) the whole LPGMZI was immersed into different RI solutions. The purpose of these experiments was to demonstrate the difference between the cases in the envelope and fringe response. Case (ii) should be immune to RI changes in the bulk solution. In both cases, the immersion region was first immersed into solution with RI of 1.3384. Then, the old solution was replaced by a new solution with a higher RI that ranged from 1.3384 to 1.3670. Before pipetting new solution into the fibre bath, the old solution was suctioned out using a suction pump (Flaem, P102P00, Desenzano del Garda, Italy) and the fibre was washed several times with DI water. The immersion period for each solution was approximately 20 min. In temperature measurements, the temperature changed uniformly over the LPGMZI from 25 °C to 30 °C with a step of 1 °C.

In IgM measurements, the surface-functionalised connecting section was immersed into IgM + Tris buffer solutions with concentration varying from 0 to 320 μg/mL. The two LPGs were left in air. For each solution, the immersion period was approximately 20 min. Then, the old solution was removed, and the immersed section was washed several times with Tris buffer solution before pipetting new solution with a higher IgM concentration into the fibre bath. Cross-sensitivity tests were conducted with IgG and IgA. The surface-functionalised connecting section was immersed into 80 μg/mL IgG, IgA, and IgM solutions sequentially. Commercial IgG and IgA were dissolved in PBS buffer solution, and commercial IgM was dissolved in Tris buffer solution. Before pipetting IgA or IgM solution into fibre bath, the old solution was suctioned off and the fibre was washed with PBS buffer solution or Tris buffer solution, respectively, as per different solvents for the antibodies. In all of the measurements, the wavelengths of fringes and envelope were investigated.

The fringes in the transmission spectrum are due to interference between cladding and core modes. The phase of the fringes relative to their envelope was determined by the difference between optical path lengths of the core and cladding modes as described by Equation (1) [17]:(1)$$\varphi =\frac{2\pi }{\lambda }\left(n_{eff}{}^{co}-n_{eff}{}^{cl}\right)L$$
where *λ* is light’s wavelength, $n_{eff}{}^{co}$ is the effective RI of core mode, $n_{eff}{}^{cl}$ is the effective RI of cladding mode, and *L* is the length of connecting section.

In this work, the wavelengths of fringe and envelope were measured and analysed in several scenarios. A discrete-Fourier transform (DFT) was applied to extract the envelope from the transmission spectrum:(2a)$$X\left[k\right]=\overset{N-1}{\underset{n=0}{\sum }}x\left[n\right]e^{-\frac{j2\pi nk}{N}}$$
(2b)$$x\left[n\right]=\frac{1}{N}\overset{N-1}{\underset{k=0}{\sum }}X\left[k\right]e^{j\frac{2\pi nk}{N}}$$
where $x\left[n\right]$ is the original signal, which is the transmission spectrum in this work, *N* is the number of samples in the signal, and $X\left[k\right]$ is the frequency domain representation of $x\left[n\right]$, where k is the wavenumber.

An example of the signal processing is shown in Figure 3. First, the DFT was applied (Equation (2a)) to the transmission spectrum (Figure 3a) to obtain its frequency domain representation (Figure 3b). The envelope and high-frequency components are distinguishable by inspection in the frequency domain. In this case, there is a local minimum at a wavenumber of 45.45 nm^−1^. These frequency components can be separated by applying low- and high-pass filters with square cut-offs at 45.45 nm^−1^ to obtain its envelope (Figure 3c1) and high-frequency components (Figure 3c2), respectively. The envelope is represented as a wide dip (~30 nm) and the high-frequency components by the narrow fringes. A simple dip-finding method was used to track the wavelengths of envelope and fringe during measurements.

### 2.2. Surface Functionalisation

#### 2.2.1. Materials

All of the chemicals used in this work were purchased from Sigma-Aldrich, Cambridge, UK. They were 3-Aminopropyltriethoxysline (APTES), sodium chloride, Tris base, sulfuric acid, peroxide solution, hydrochloric acid (37%), ethanol, glutaric dialdehyde solution (glutaraldehyde, 50% in H_2_O), IgM, IgG, and IgA from human serum, and anti-human IgM antibody produced in goat.

#### 2.2.2. Preparation Scheme

The connecting section of a bare LPGMZI was first immersed in Piranha solution for 20 min to remove the organic contamination and generate hydroxyl groups on the surface. Then, it was washed with DI water and dried with nitrogen. Secondly, the connecting section was immersed in APTES solution (2 *v/v*% in ethanol) for 20 min. Then, it was washed with DI water and ethanol alternately and dried in normal environment overnight (~16 h). Thirdly, the connecting section was immersed in glutaric dialdehyde solution (50 wt% in H_2_O) for 2 h followed by washing with DI water. Fourthly, the connecting section was immersed in 1 mg/mL anti-IgM solution for 2 h followed by washing with PBS buffer solution. The surface functionalisation process is shown in Figure 4a. The glutaraldehyde works as cross-linker that reacts with amine groups (Figure 4b).

## 3. Results

### 3.1. Transmission Spectra

An example of the transmission spectra of an LPGMZI when the connecting section is immersed in different solutions is shown in Figure 5. There are multiple attenuation bands in the transmission spectrum (Figure 5a). The attenuation band located between 880 nm and 940 nm is relatively wider and the fringes have a higher contrast. This makes the change in signal easier to analyse, so this is utilised throughout. As shown in Figure 5b, there are six significant dips (<0.8) in the transmission spectrum. As the concentration of IgM increases, the spectrum shifts, the envelope remains static, and the relative depth of each dip changes. From Figure 5c, the wavelength shift of a fringe can be observed more clearly.

### 3.2. Temperature Measurements

The LPGMZI’s temperature measurements are shown in Figure 6. Due to the experimental configuration described in Section 2.1, the temperature changes uniformly over the whole LPGMZI. As shown in Figure 6a, both the envelope and fringes’ wavelengths change with temperature. Figure 6b shows the difference between them. The signal remains almost constant at zero, which means that the envelope and fringe’s wavelengths change with temperature at the same rate. The fluctuations in Figure 6b between each temperature step is due to the heating process. The temperature in the bath is not uniform during the heating process. Figure 6c gives the calibration results of the envelope’s wavelength shift, exhibiting a sensitivity of 0.3341 nm/°C.

### 3.3. External Refractive Index Measurements

Figure 7a shows the results when the external RI outside the connecting section changes and the two LPGs remain in air. The effective RI of the cladding mode increases with an increased external RI, which leads to an increase in the optical path length and a shift in the fringes. The wavelength of the envelope remains constant in this case as the external conditions (temperature, RI) of the two LPGs do not change. Figure 7b gives the difference between the fringe and envelope’s wavelength shifts. From the results in Section 3.2, the interference from temperature can be compensated for by calculating the difference. Figure 7c shows the results when the external RI outside the whole LPGMZI changes (connecting section and two LPGs). The wavelengths of the fringe and envelope change simultaneously at the same rate. Figure 7d shows the difference between the wavelengths of fringes and the envelope. When the external RI over the whole LPGMZI changes, the difference is close to zero. This demonstrates that the fringe and envelope respond to the external RI with the same sensitivity. The slight difference is most likely caused by the imperfection of signal processing as the envelope and high-frequency component overlap slightly in the frequency domain. Figure 7e shows the calibration curves of the fringe’s wavelength shifts in these two scenarios. The fringes’ wavelength changes linearly with the external RI in both scenarios and the sensitivity is slightly higher when the whole LPGMZI is immersed. The spikes in Figure 7a–d are due to the change in solution and temporary exposure of the sensing region to air.

### 3.4. IgM Measuremnts

The results of the IgM measurement are shown in Figure 8. Only the connecting section is immersed in IgM solutions. This is selected due to the ease of experimentation as the bulk RI of the buffer solution does not change significantly in comparison to IgM measurements (Appendix A). As shown in Figure 8a, the fringes’ wavelength changes with IgM concentration whereas the envelope’s wavelength remains almost constant. The small fluctuations in the envelope’s wavelength are most likely caused by temperature fluctuations in the laboratory environment. The difference between the fringes’ and envelope’s wavelengths was calculated to compensate for the signal’s cross-sensitivity to temperature (Figure 8b). The spikes in Figure 8a,b are due to the change in solution and temporary exposure of the sensing region to air.

After visual inspection of the data, a second-order polynomial fitting was applied to obtain the calibration curve (Figure 8c), with a coefficient of determination of 99.17%. To investigate the temperature cross-sensitivity, an 80μg/mL IgM solution was tested under three different temperatures (Figure 8d). The response reduces slightly with an increasing temperature [18], but this is expected as a higher temperature degrades the intrinsic affinity of antibodies, which leads to a lower binding between the antibody and bioreceptor [18].

Figure 8e shows the results of the sensor’s selectivity test. The signal remains low in the 80 μg/mL IgA or IgG solution and increases in the 80 μg/mL IgM solution. The selectivity can be quantitively assessed by Equation (3) [19].
(3)$$S=\frac{R_{IgM}}{R_{I}}$$
where $R_{IgM}$ is the sensor’s response to IgM and $R_{I}$ is the sensor’s response to the interfering analyte (IgA or IgG). In this work, the selectivity is *S* = 17.9.

The limit of detection (LoD) can be calculated according to Equation (4) [20]
(4)$$\mathrm{LoD}=\frac{3S_{y}}{S}$$
where *Sy* is the standard deviation of the response at baseline and *S* is the first-order derivative of the calibration curve at zero concentration.

## 4. Discussion

Many immunosensors are based on the detection of the RI change on the sensor’s surface, such as LPG-based sensors and surface plasmon resonance (SPR)-based sensors. In these cases, the bulk RI of the measured solution might change, which introduces an unwanted change to the measured signal. According to the results from Section 3.1, the LPGMZI’s fringes and envelope have almost the same wavelength shifting rate with the external RI if the whole LPGMZI is immersed. The difference between the wavelength shifts of the envelope and fringe is not sensitive to bulk RIs (Figure 7d) in this case. This property can be utilised to compensate for the cross-sensitivity to bulk RI change. The whole LPGMZI can be immersed into antibody solutions but only the connecting section is designed to be sensitive to antibodies. However, in practice, the non-specific binding of antibodies to LPGs can also cause a wavelength shift of the envelope, which affects the measurement of bulk RIs [21,22]. The non-specific binding of IgM to an LPG is shown in the Appendix A). A measurement of an 80 μg/mL IgM solution with the whole LPGMZI immersed was performed (Appendix A). The connecting section was functionalised with anti-IgM. The wavelength shift of the envelope is caused by the non-specific binding of IgM to bare LPGs. The fringe has a greater wavelength shift than the envelope because the functionalised connecting section has a higher affinity to IgM than bare LPGs. The non-specific binding of IgM to bare LPGs can affect the measurement of bulk RIs. As a result, the IgM sensor developed in this work can only compensate for the cross-sensitivity to temperature fluctuations. Further work needs to be carried out to prevent non-specific binding, such as depositing a poly(ethylene glycol) (PEG) film on the surface of the LPGs [21]. The characteristics of relevant optical IgM sensors described in the literature are listed in Table 1.

The sensitivity of the IgM sensor developed in this work is lower than other single-LPG-based IgM sensors [3,23]. One cause is that the LPGs constituting the LPGMZI are not operating at the phase matching condition, in contrast to the other two single-LPG-based IgM sensors [3,23]. One way to improve the sensitivity is by fabricating the LPGs to work closer to the phase matching condition. In this case, the analysis will be more complex as the transmission spectrum includes not only the wavelength shift but also the merging and separation of attenuation bands [3]. A second approach is to increase the surface coverage of bioreceptors, and to utilise silica nanoparticles (SiNPs) and gold nanoparticles (AuNPs) [3].

The selectivity of the designed IgM sensor is higher than the sensors developed in other work: approximately 90% higher than the single-LPG-based sensor [23], 12% higher than the LSPR-based sensor [24], and approximately 26% higher than the WGM microsphere-based sensor [25]. The high selectivity is most likely to be due to contributions by the surface functionalisation method. Compared with the single-LPG-based sensor, the sensor designed in this work has the same substrate. However, poly (allylamine hydrochloride, PAH) and silica nanoparticles (SiNPs) dual layers are assembled on the surface of a single LPG [23], which potentially provides more non-specific binding sites. This work utilised a similar surface functionalisation scheme to the described LSPR-based sensor [24]. Both have glutaraldehyde as a cross-linker, but the amine groups were generated on the gold surface of the LSPR platform rather than silica surface [24]. This might degrade the quality of the surface functionalisation. The WGM-microsphere-based sensor has amine groups on a silica surface [25]. However, the bio-receptors are immobilised by physical adsorption without any cross-linkers [26], which might lead to a lower surface coverage of bio-receptors and the exposure of more non-specific binding sites [27].

Real samples, such as human serum, contain multiple biomolecules. The high selectivity of the designed sensor is advantageous in practical applications. The selectivity test was conducted with IgG and IgA. The commercial IgG and IgA were dissolved in PBS buffer solution (product specification), but the commercial IgM was dissolved in Tris buffer solution. However, the LPG has almost the same baseline in these two buffer solutions (Appendix A).

Individual sensors have been found to be repeatable and stable, as demonstrated in an experiment in which the connecting section of the sensor was immersed in Tris buffer solution three times (Appendix A) and a standard deviation of 0.0055 nm was obtained. However, in order to achieve a repeatable commercial sensor, more development work is required in the manufacturing process, such as accurately inscribing the grating configuration and obtaining uniform coatings.

## 5. Conclusions

In this work, an LPGMZI immunosensor was fabricated and characterised. Due to the interference between cladding and core modes, fringes are present in the transmission spectrum. The LPGMZI configuration was tested for the cross-sensitivity to temperature and bulk RIs. For the response to bulk RIs, if only the connecting section is immersed, the wavelengths of the fringes shift with an external RI whereas the wavelength of the envelope remains constant. If the whole LPGMZI is immersed, the wavelengths of the fringes and envelope shift together with an external RI at the same rate. For the response to temperature, the wavelengths of the fringes and envelope shift together at the same rate with a homogeneous temperature change.

For immuno-sensing, an anti-human IgM antibody was deposited on the connecting section of the LPGMZI. When only the connecting section is immersed into IgM solutions, the wavelengths of the fringes shift with IgM concentration whereas the wavelength of the envelope remains almost constant at different IgM concentrations. In comparison to the IgM sensors designed based on single LPGs, the LPGMZI-based IgM sensor designed in this work has a lower sensitivity. However, it provides a higher selectivity due to the surface functionalisation method and can compensate for cross-sensitivity. These advantages provide it with the potential to be a robust portable immuno-sensing platform. This platform is versatile as, by changing the functional layer, there is potential to measure a range of other analytes, such as IgG, λ chain, and κ chain proteins.

## Figures and Tables

**Figure 1 biosensors-12-01099-f001:**
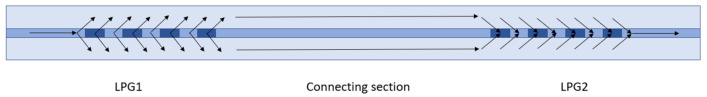
LPGMZI structure.

**Figure 2 biosensors-12-01099-f002:**
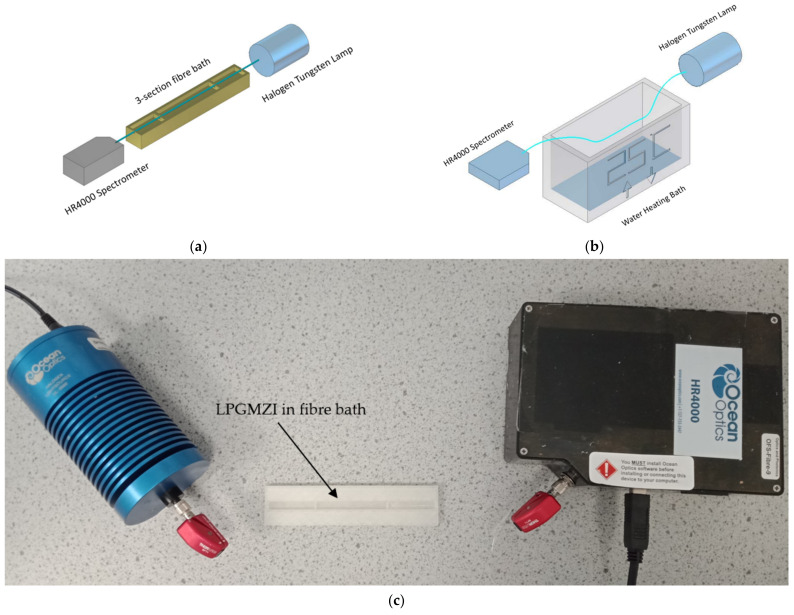
The experimental set-ups: (**a**) the LPGMZI in a 3-section fibre bath used for external RI measurement and IgM measurement (LPG sections 2.75 cm (L) × 0.5 cm (W) × 0.3 cm (H), middle section 4.5 cm (L) × 0.5 cm (W) × 0.3 cm (H); (**b**) the LPGMZI in a water heating bath used for temperature measurement (30 cm (L) × 15 cm (W) × 15 cm (H)); (**c**) the photo of set-up.

**Figure 3 biosensors-12-01099-f003:**
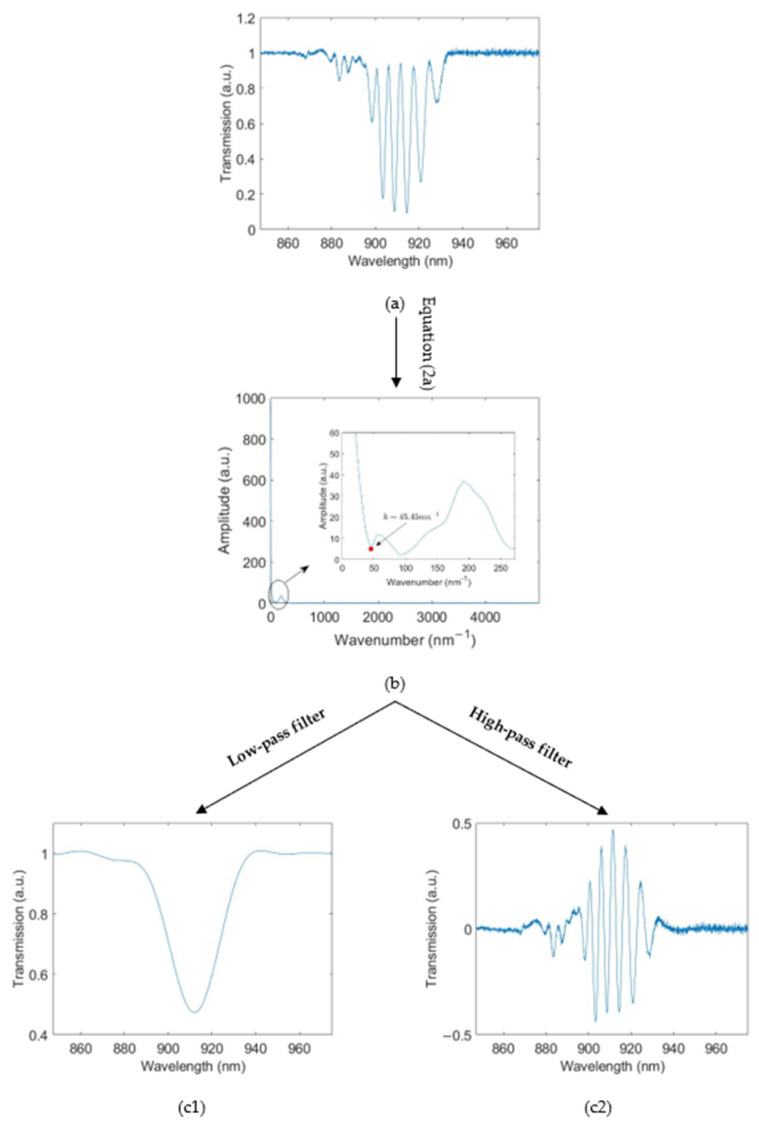
Signal processing: (**a**) transmission spectrum; (**b**) frequency domain representation of the transmission spectrum; (**c1**) envelope; (**c2**) high−frequency components. The wavenumber calculation is shown in the Appendix A.

**Figure 4 biosensors-12-01099-f004:**
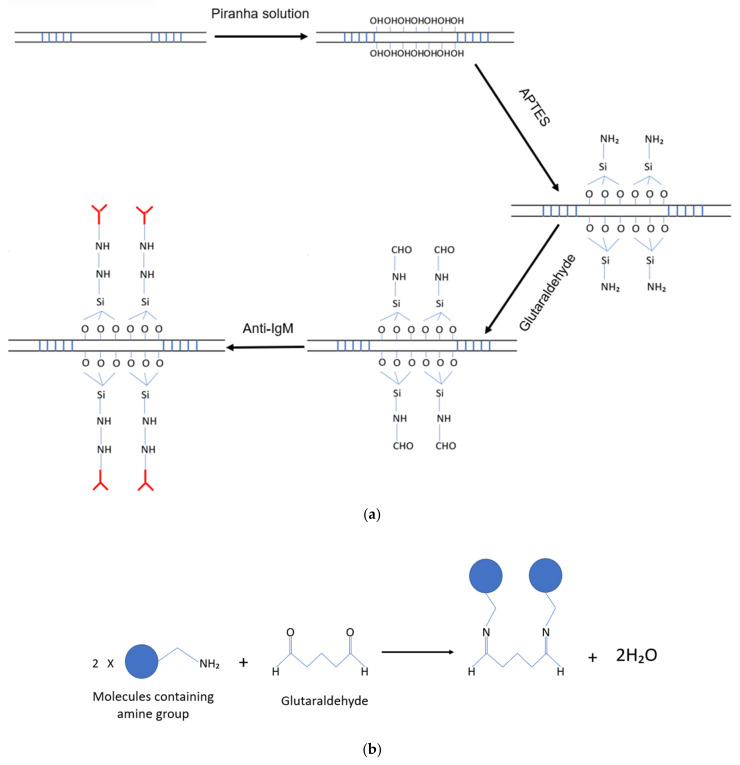
(**a**) The surface functionalisation of LPGMZI. (**b**) Glutaraldehyde works as cross-linker.

**Figure 5 biosensors-12-01099-f005:**
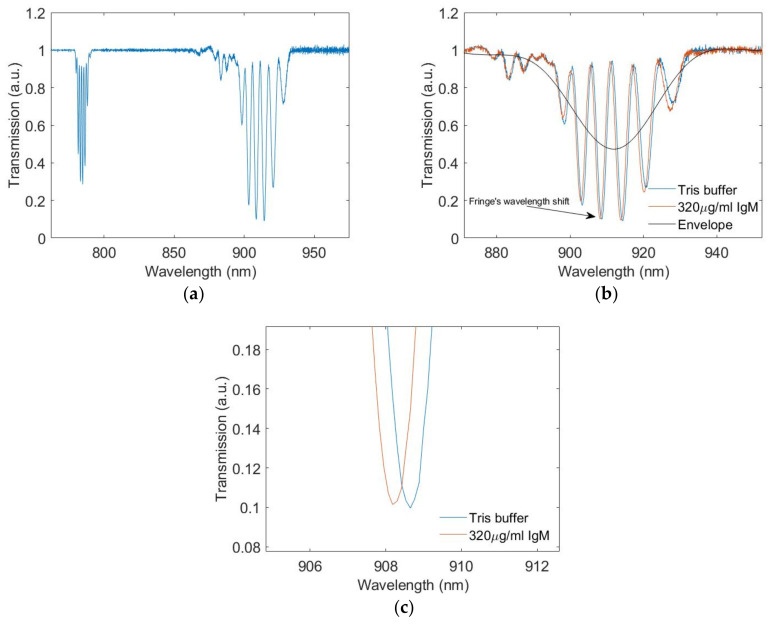
(**a**) Comparison between different attenuation bands in the transmission spectrum. (**b**) The transmission spectra of the LPGMZI when the connecting section is immersed in solutions with different IgM concentrations. (**c**) The wavelength shift of a fringe during the measurement.

**Figure 6 biosensors-12-01099-f006:**
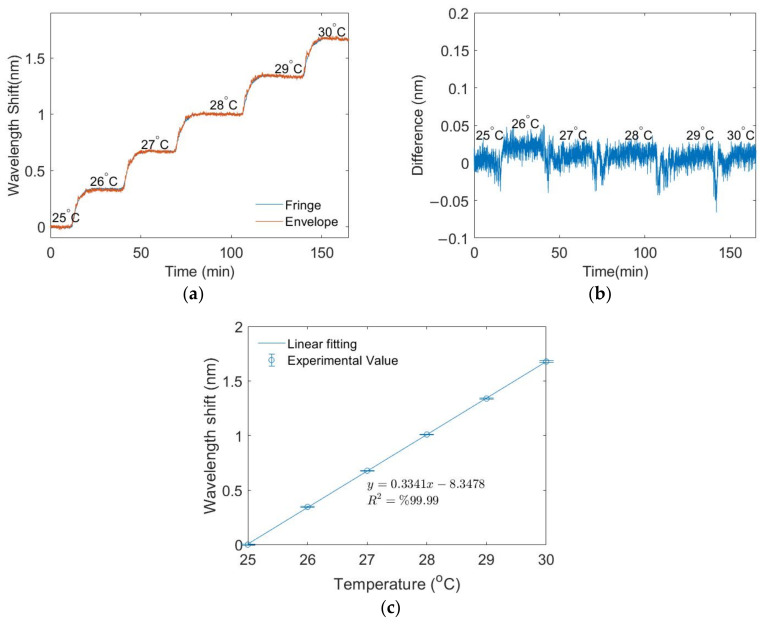
Temperature measurements results: (**a**) wavelength shifts of fringe and envelope; (**b**) difference between the wavelength shifts of fringes and envelope; (**c**) calibration curve of the envelope’s wavelength shift.

**Figure 7 biosensors-12-01099-f007:**
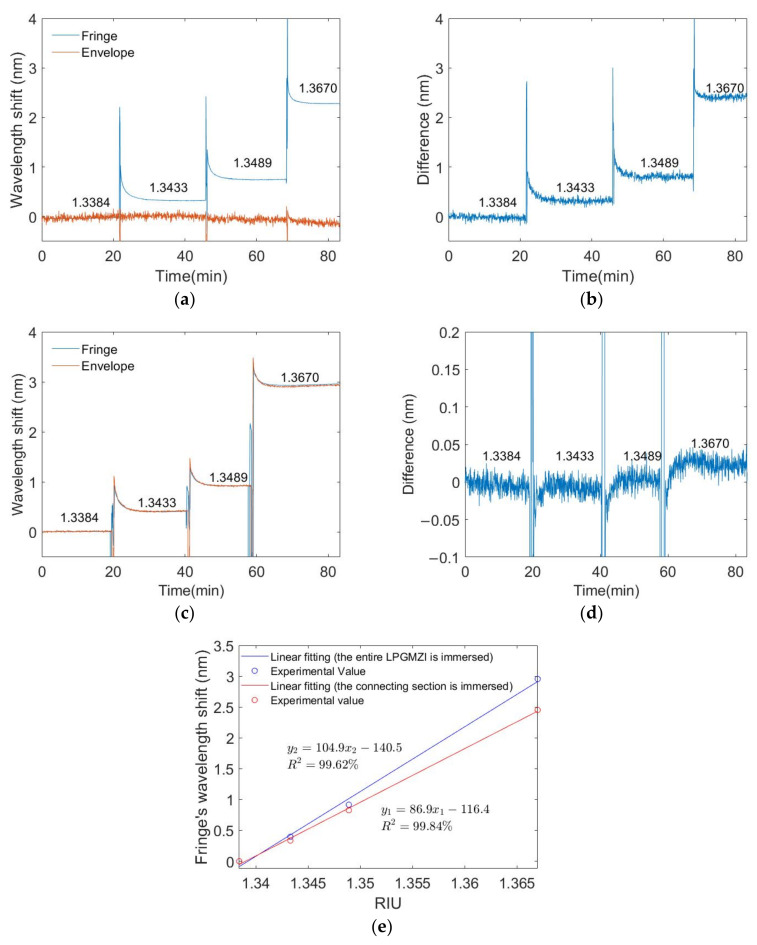
External RI measurements: (**a**) Wavelength shifts of fringes and envelope when the connecting section is immersed in RI solutions and (**b**) their difference. (**c**) Wavelength shifts of fringes and envelope when the entire LPGMZI is immersed in RI solutions and (**d**) their difference. (**e**) Calibration curve of the fringes’ wavelength shifts in those two different cases.

**Figure 8 biosensors-12-01099-f008:**
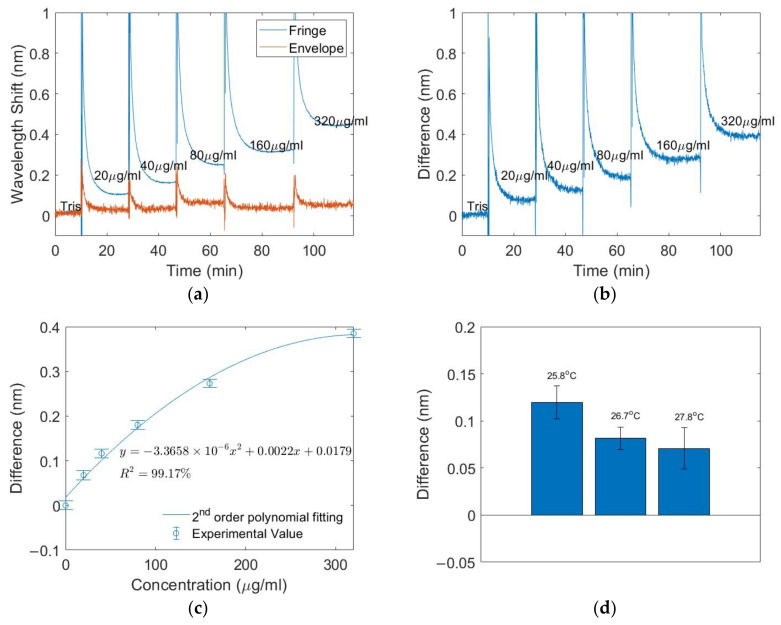
IgM measurements results: (**a**) Wavelength shift of fringe and envelope. (**b**) Difference between the wavelength shifts of fringes and envelope. (**c**) Calibration curve of difference versus concentration. (**d**) Response to 80 μg/mL IgM solution under different temperatures. (**e**) Selectivity test.

**Table 1 biosensors-12-01099-t001:** The characteristics of different optical IgM sensors.

Platform	Measurand	IgM Concentration Range	Signal Change at the Highest Concentration	Limit of Detection	Selectivity	Reference
Long period grating	Wavelength shift	15.6 μg/mL–1 mg/mL	~12 nm	15 pg/mm^2^	Not stated	[3]
Long period grating	Wavelength shift	10 μg/mL–320 μ/mL	4.96 nm	Not Stated	9.4	[23]
LSPR on substrate	Wavelength shift	1 μg/mL–100 μg/mL	~1.6 nm	1 μg/mL	~16	[24]
WGM microsphere	Wavelength shift	1 μg/mL–100 μg/mL	~1.7 nm	2 μg/mL	~14.2	[25]
LPGMZI	Wavelength shift	20 μg/mL-320 μg/mL	0.39 nm	13 μg/mL	17.9	This work

## Data Availability

The data presented in this study are available on request from the corresponding author.

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
