# Peer review of "Long Period Grating Mach–Zehnder Interferometer Based Immunosensor with Temperature and Bulk Refractive Index Compensation"

_biosensors, 2022, doi:10.3390/bios12121099_

Round 1

Reviewer 1 Report

This study deals with development and demonstration of LPG based MZI for IgM detection. The sensor has two LPGs and a connecting section in between them where the anti-IgM was immobilized. Authors obtained the shift in wavelength and envelope through DFT of the spectrometer output. They systematically investigated the influence of temperature and bulk RI changes on the sensor response under the two conditions, when (1) the connecting section alone and (2) entire region including 2 LPG section were exposed to the sensing medium. Its interesting to observe their results showing a distinguishable frequency/wavlength shift for IgM binding and buffer. The following points may be considered to improve the clarity and presentation.

1. It is not clear how the authors obtained Fig. 3b with x-axis title of 9.09 nm-1. An detailed explanation may be added for the same and the subsequent process.

2. Fig. 6b, the y-axis may be zoomed to highlight the difference. What is the standard deviation in the Fig. 6c and Fig. 8c.

3. What are the reasons for the spikes in the response shown in Fig. 7 and 8.

4. The sensitivity of the sensor seems to be low as of now as already acknowledged by the authors. How to improve the same may be further highlighted.

Reviewer 2 Report

1.The way of expression like In [12] had better be modified in several paragraphs beginning from line 36.

2.Characters of waveform are suggested to be summarized and highlighted in Figure 3c. 

3.More prospects are suggested to described in the section Conclusions.

Reviewer 3 Report

This paper focuses on immuno-sensors based on a long period grating Mach-Zehnder interferometer configuration. The authors propose and demonstrate a new signal processing technique to achieve improved immunity to bulk refractive index changes and temperature changes. Although this structure has been quite extensively investigated in the past, the new method appears to be novel and quite successful at reducing the impact of environmental factors on measurement performance. The authors reference previous work in this area quite widely and the experiments appear to have been conducted carefully. However I do have several comments and suggestions to improve the clarity and impact of this paper:

1. The title does not adequately capture the novelty of the paper; I suggest that you include 'with improved insensitivity to temperature and refractive index drift' or something similar

2. Equation 2 is linked to reference 12, but in that paper the equation is not derived, merely cited. Readers should be referred to the original derivation, which is in Y Liu, J.A.R Williams, L Zhang, I Bennion, Phase shifted and cascaded long-period fiber gratings, Optics Communications, Volume 164, Issues 1–3, 1999, Pages 27-31

3. The parameter 'k' should be defined in equation 3(a) (presumably wavenumber)

4. The x-axis in Figure 3(b) should have a proper label (wavenumber again, or spatial frequency?); the current '9.09 nm^-1' label is confusing.

5. The signal processing method that you use is not sufficiently explained. How are the low-pass and high-pass filters defined? Are they square filters, or shaped? How are the cut-on/cut-off frequencies determined? The reader needs to be able to replicate your results and that is impossible with the information provided.

6. How are the wavelength shifts tracked in the experimental results? Is a simple dip-finding method used or is a more sophisticated method applied?

7. For the curve fitting for Fig 8(c) is there a theoretical reason to expect that a 2nd order polynomial would provide a good fit, or is this simply from inspection?

8. In section 3.4 the limit of detection should be reported. This would allow the method to be better compared with other immunosensors.

9. Whilst it is good to see that the selectivity was found to be good, it appears that this is a function of the surface functionalisation only, and is not connected to the new signal processing method. Since this paper is presenting advances in signal processing, it would be good to clarify that.

10. The quality of the written English could be improved significantly. 

Reviewer 4 Report

The work presented by Korposh and co-workers entitled “Long Period Grating Mach-Zehnder Interferometer Based Immunosensor” describes the fabrication of a Long Period Grating Mach-Zehnder Interferometer, its functionalization with anti-IgM and its application for the selective detection of IgM versus IgA and IgG. They have designed a 3-section fibre bath which allows the analysis of the gratings and the connection section separately. Then, they have analysed the fringes and the envelop of the spectra produced in the different experiment tested (temperature fluctuation, IgM detection and RI changes) and have observed that both the fringes and envelop change with the different conditions when the entire fibre is immersed in the testing solution while only the fringes shift when the connecting zone is immersed in the testing solution.

They stated that the design of this immunosensor, experimental set-ups and signal processing has the potential to compensate for the temperature and bulk RI fluctuations that can be observed when LPG platforms are used for sensing.

The general idea of the paper offers great significance in the field, the manuscript is well written and organised and the described experiments are good. However, the final results weakly support the conclusions.

Therefore, major revisions are recommended before accepting the paper for its publication in Biosensors:

1.       Page 11, Section 3.4.

·       IgG measurements are carried out with only the connecting zone functionalized with anti-IgM immersed in solutions of increasing concentrations of IgG. A good correlation between the wavelength shifts in the fringes and the IgG concentrations is observed. Small changes in the envelop are associated to temperature fluctuations in the laboratory environment as, according to their analysis, the envelop signal should remain constant if only the connecting zone is immersed. Although the hypothesis about the temperature is based on previous results and it is likely to be confirmed, it does not sufficiently prove the final conclusions. Indeed, the small wavelength shifts around 0.1 nm in the envelop signal in figure 8a are associated to changes below 1ºC in figure 6a. Higher changes of temperature would yield wavelength shifts higher than the signal of the highest concentration of IgG; for example, 2 ºC provokes 0.7 nm shifts in figure 6a, higher than the 0.5 nm shifts observed with 320 µg/mL of IgG in figure 8a). Therefore, IgG measurements at different environmental temperatures should be carried out to better support the conclusion about the robustness of the platform to temperature. At least, 3 different temperatures between 25 and 30 ºC should be assessed.

·       The same discussion applies for the influence of the RI environment on the results in the IgG measurements. First, authors should clearly indicate the medium in which the gratings of the fiber are immersed while they are analyzing the connecting zone in the medium section of the bath. Then, if the authors want to prove that effect of RI environment does not affect to the IgG detection, measurements of the IgG in different RI environments should be carried out.

·       Controls with the effect of the different buffers could also be included in figure 8d, as every immunoglobulin is assessed in a different buffer solution.

·       For gaining consistency, experiments should be repeated at least three times and the standard deviation of the obtained values should be included in the results.

Minor revisions are also recommended and include:

1.     Page 7, section 2.2.2 and figure 4. The authors describe in the abstract and conclusions that only the connecting zone of the fibre is functionalised with the anti-IgM antibodies. However, this is not clear in section 2.2.2, neither in the text nor in the scheme of figure 4a. Please, change text of section 2.2.2 and scheme of figure 4a, so they clearly indicate that only the connecting zone is functionalised.

2.     Page 13, line 321. The authors hypothesize that the direct immobilization of the bioreceptors through silane coupling can provoke higher non-specific binding that the functionalization using crosslinkers such as glutaraldehyde. According to the reviewer experience this is not totally true and glutaraldehyde coupling can lead to undesired binging events. So, please, remove this sentence or include some bibliography that can support your hypothesis.

Reviewer 5 Report

In this paper “Long Period Grating Mach-Zehnder Interferometer Based Immunosensor". The authors proposed a long period grating Mach-Zehnder Interferometer (LPGMZI) that consists of two identical long period gratings (LPGs) in a single fibre to measure immunoglobulin M (IgM). The approach is interesting however, there are several points to be clarified to make this paper publishable in this journal.

1. Author is suggested to improve introduction as it is too long, it is closer to a popular science articles rather than a section in a research paper. Please shorten and simplify it in the revision.

2. Author is suggested to improve figures quality.

3. Is it possible for author to provide optimization data for non-specific adsorption on the surface?

4. The authors should provide more discussion on the selectivity and stability of the fabricated sensor.

5. It is clearly mentioned in the paper that sensitivity of proposed sensor is less than previously reported sensor which is a drawback of this work. Can author provide a possible solution to overcome this drawback?

6. Author characterized sensor in two different scenarios (a) when connecting section is immersed in RI solution and (b) when the entire LPGMZI is immersed in RI solutions but from calibration curve only slight difference in sensitivity is observed. The purpose of using two different scenarios is not well understood, author should provide proper explanation for this.

7. To show repeatability of LPGMZI, can author add error bar in the results?

8. Can author provide approximate cost and optical image of proposed sensor?

9. Author is requested to revise manuscript carefully as there are some typo errors in the manuscript.

Round 2

Reviewer 4 Report

The comments previously suggested have been addressed. Therefore, I recommend the publication of this articles in the present form. 

Reviewer 5 Report

The reviewers have addressed my concerns. I recommend this paper for publication.